# Antimicrobial Strategies Proposed for the Treatment of *S. pseudintermedius* and Other Dermato-Pathogenic *Staphylococcus* spp. in Companion Animals: A Narrative Review

**DOI:** 10.3390/vetsci11070311

**Published:** 2024-07-11

**Authors:** Valentina Stefanetti, Fabrizio Passamonti, Elisa Rampacci

**Affiliations:** 1Department of Human Science and Promotion of Quality Life, San Raffaele Telematic University, 00166 Rome, Italy; valentina.stefanetti@collaboratori.unipg.it; 2Department of Veterinary Medicine, University of Perugia, Via San Costanzo 4, 06126 Perugia, Italy; elisa.rampacci@unipg.it

**Keywords:** *Staphylococcus* spp., *Staphylococcus pseudintermedius*, plant antibacterial activity, new antimicrobial compounds, skin antiseptics, bacteriophages, adjuvants

## Abstract

**Simple Summary:**

*Staphylococcus* skin infections in companion animals are a growing concern due to their impact on animal health and the risk of spreading these infections to humans. Traditional antibiotics are becoming less effective because bacteria are developing resistance to them, underlining the need for new alternative treatment approaches to prevent and control staphylococcal infections. This review explores new strategies to fight these infections, including novel antimicrobials, repurposing existing drugs for new uses, natural products, skin antiseptics, and other new treatments. These strategies can help maintain the health of companion animals, reduce the risk of transmission to people, and ultimately benefit public health by limiting the spread of resistant bacteria.

**Abstract:**

The treatment of dermato-pathogenic *Staphylococcus* spp., particularly *Staphylococcus pseudintermedius*, in companion animals presents significant challenges due to rising antimicrobial resistance. This review explores innovative strategies to combat these infections. We examined novel antimicrobials and the repurposing of existing drugs to enhance their efficacy against resistant strains. Additionally, we evaluate the potential of natural products, nanomaterials, and skin antiseptics as alternative treatments. The review also investigates the use of antimicrobial peptides and bacteriophages, highlighting their targeted action against staphylococcal pathogens. Furthermore, the role of adjuvants in antibiotic treatments, such as antimicrobial resistance breakers, is discussed, emphasizing their ability to enhance therapeutic outcomes. Our analysis underscores the importance of a multifaceted approach in developing effective antimicrobial strategies for companion animals, aiming to mitigate resistance and improve clinical management of staphylococcal skin infections.

## 1. Introduction

Skin infections are common in companion animals, leading to a range of dermatological issues, with an impact on the skin surface microenvironment that promotes bacterial multiplication. Different staphylococcal species are responsible for feline and canine cutaneous infections [1]. Particularly, *Staphylococcus pseudintermedius* (*S. pseudintermedius*) is considered the main causative agent of canine pyoderma, otitis externa, and wound infections in dogs [1], and it is also able to produce biofilm [2]. In recent years, methicillin-resistant *S. pseudintermedius* strains (MRSP) and their increasing resistance to conventional antibiotics have emerged in companion animals worldwide, often complicating treatment options and control measures [3,4]. To contrast this phenomenon, current guidelines suggest that antimicrobial prescribing has to be based on laboratory antimicrobial susceptibility testing (AST), although some interpretative bias associated with the use of automatic AST on *S. pseudintermedius* was reported [4].

In this scenario, it is crucial to consider alternative treatments and to develop new strategies to manage staphylococcal skin infections in companion animals. This review explores various innovative approaches, including novel antimicrobials, repurposed drugs, natural products, nanomaterials, skin antiseptics, antimicrobial peptides, bacteriophages, and adjuvants (Figure 1). By evaluating these alternative strategies, we aim to provide a comprehensive overview of potential solutions to improve the clinical management of staphylococcal skin infections in companion animals and to address the growing issue of antimicrobial resistance.

## 2. Conventional Antibiotics

General guidelines for the antimicrobial therapy of cutaneous infections of companion animals are available in the literature, including topic and systemic approaches [5]. Antibiotic classes for systemic treatment include β-lactams, lincosamides, sulfonamides, tetracyclines, fluoroquinolones, aminoglycoside, phenols, and rifamycins. Their suggested clinical dose and therapeutic regimen are reported in a previous review [5]. The selection priority of the antibiotic compound must be based on (a) the results of AST, (b) the consideration of the potential toxicity of some molecules, and (c) prudence principles of antibiotic use to preserve critically important antibiotics for human use, as recommended by the European Medicines Agency (EMA) based on the European regulations on the use of antibiotics in veterinary medicine [6]. Alongside this, researchers investigated the killing activity of different antibiotics on varied bacterial densities of *S. pseudintermedius* based on the rationale that in vivo bacterial concentrations are supposed to exceed that used in susceptibility testing. Doxycycline exhibited bacteriostatic properties regardless of bacterial densities and/or drug concentrations tested, with a minor killing capability compared to cefazolin, cefovecin, and pradofloxacin. CLSI recommends specific interpretative breakpoints of doxycycline for canine *S. pseudintermedius* that differ from the criteria for human isolates. In this regard, interpretive bias could be associated with the use of automatic AST methods that may include concentration range for doxycycline that do not match CLSI clinical breakpoints for canine *S. pseudintermedius* [4]. The fluoroquinolone pradofloxacin required short drug exposure times, while cefazolin and cefovecin were evaluated as time-dependent antibiotics [7]. Fluoroquinolones enrofloxacin and its active metabolite ciprofloxacin demonstrated a rather rapid killing against *S. pseudintermedius* as well [8]. These findings may influence the time of clinical resolution and thus the duration of therapy. Based on pharmacokinetic profiles, the use of fluoroquinolones, including enrofloxacin, marbofloxacin, and pradofloxacin, is considered to be useful in the management of pyoderma in companion animals, including deep pyoderma and extensive lesions [9,10]. However, the European categorization of drug priority has to be observed. Clindamycin, a lincosamide drug, is in EMA category C (caution use); based on that, it is one of the most common antibiotics used for skin infections in companion animals. In this regard, a previous study supported the administration of clindamycin at the dose of 11 mg/kg, once daily, rather than 5.5 mg/kg, twice daily, for the treatment of canine pyoderma caused by *S. pseudintermedius* [11]. It is not clear if other tetracyclines might have better killing activity than doxycycline. Minocycline could be considered a treatment option for some doxycycline-resistant *S. pseudintermedius* because some genetic mechanisms confer resistance to doxycycline but not minocycline [12]. Macrolides are not usually administered for the treatment of dermato-pathogenic *Staphylococcus* spp. in companion animals. Clarithromycin demonstrated anti-biofilm properties against *S. aureus;* on the contrary, it did not inhibit biofilm formation by *S. pseudintermedius* on polystyrene [13]. Rifampicin has been tested on *S. pseudintermedius* strains, and it was able to inhibit biofilm formation in all the strains but was less effective in biofilm eradication [14]. However, rifampicin should not be selected as a first-choice antibiotic, being considered a critically important molecule for human treatment, particularly in the case of mycobacterial infections.

Over the years, researchers also investigated the comparative efficacy of several antibiotic combinations against dermato-pathogenic *Staphylococcus* spp. Several combinations of antibiotic/antifungal/corticosteroid are available in the market and are considered the standard of care for canine otitis externa based on ear cleaning and administration of the combination once or twice daily for 5–7 days at home. However, this therapeutic protocol should be reviewed. It was indeed demonstrated that specific combinations, such as terbinafine/florfenicol/betamethasone administered twice at a one-week interval, are an effective and convenient approach to treating mixed infections of *S. pseudintermedius* and yeasts [15]. An otic suspension containing gentamicin/posaconazole/mometasone furoate was also developed as a single-dose product that proved to be equally effective to two administrations of terbinafine/florfenicol/betamethasone [16]. Gentamicin was impregnated in polymethylmethacrylate (PMMA) beads by Morrison et al. alone and in combination with silver [17]. The authors demonstrated that gentamicin-impregnated PMMA was effective in reducing biofilm produced by gentamicin-susceptible *S. pseudintermedius*, while it had no effect on gentamicin-resistant isolates. This paper addresses the necessity of making advances in pharmaceutical technology applied to the veterinary sector, so far lacking in the scientific literature. Finally, it is noteworthy to mention interactions of last resort antibiotics such as the triple β-lactam combination meropenem/piperacillin/tazobactam that demonstrated broad antimicrobial coverage against various resistance phenotypes of *S. aureus*, *S. epidermidis*, and *S. pseudintermedius* [18]. Gonzales et al. reported a strong synergistic activity of these compounds in a 1:1:1 ratio and the ability to suppress resistance evolvability [19]. However, meropenem and the combination piperacillin/tazobactam are listed in the EMA category A, including drugs that may be given to companion animals only under exceptional circumstances.

## 3. Novel-Purposed Antimicrobials and Drug Repurposing

The urgent demand for the development of novel antimicrobial therapies has introduced novel strategies to be potentially used for the treatment of skin pathogenic staphylococci. Among novel-purposed compounds, enzymes of the methylerythritol phosphate pathway (MEP) represent potential targets for the treatment of *S. pseudintermedius* and other dermato-pathogenic staphylococci. In this context, **fosmidomycin** inhibits the deoxyxylulose phosphate reductoisomerase, an enzyme that catalyzes the first step of the isoprenoid biosynthesis in staphylococci, including *S. schleiferi* and *S. pseudintermedius* [20]. In 1978, fosmidomycin was first described as a new class of antibiotics isolated from *Streptomyces lavendulae* and *Streptomyces rubellomurinus* [21,22]. Ester modification of fosmidomycin led to the synthesis of lipophilic prodrug analogs with enhanced anti-staphylococcal potency associated with improved PK/PD properties. Ester modification indeed facilitated the entry of the MEP inhibitors inside bacterial cells by avoiding the dependency on active GlpT-mediated transport [20].

Mattio et al. synthesized a **3-decyltetramic acid analog of the ureido dipeptide natural antibiotic leopolic acid A** that showed a more effective antimicrobial activity than the parent leopolic acid against *S. pseudintermedius*, including multidrug-resistant strains [23]. As shown in Table 1, the MIC_50_ of the leopolic-acid-inspired compound was 8 μg/mL against methicillin-susceptible *S. pseudintermedius* (MSSP) and 32 μg/mL against multidrug-resistant phenotypes.

Drug repurposing is a cost-effective strategy based on the assessment of existing approved drugs for new therapeutic purposes. This paragraph reported pharmacological compounds proposed as antimicrobial therapies for animal dermato-pathogenic *Staphylococcus* spp.

**Antiparasitic drugs**, particularly the anthelmintic drug *oxyclozanide*, fasciolicide *triclabendazole*, and anticoccidial *ionophores*, were tested against methicillin-susceptible and methicillin-resistant *S. aureus* and *S. pseudintermedius* [24,25,26,27]. Their MIC values are displayed in Table 1. *Lasalocid, monensin, narasin*, and *salinomycin*, which are anticoccidial cationic polyether ionophores, showed appreciable antimicrobial activity against both methicillin-susceptible (MSSA) and methicillin-resistant (MRSA) *S. aureus* (Table 1). Particularly, narasin demonstrated the most potent activity on *S. aureus*, also confirmed on MSSP and MRSP (MIC ranged between 0.06–0.25 μg/mL) [26,27]. Moreover, 4–16 μg/mL of narasin significantly reduced S. *pseudintermedius* biofilms, but its minimum biofilm eradication concentration (MBEC) was not measured [28]. *Oxyclozanide* showed low MIC against MSSP, MSSA, and MRSP and a Mutant Prevention Concentration (MPC), which is a threshold above which the selective proliferation of resistant mutants is expected to occur only rarely, ranging from 16 to 32 μg/mL [24]. The MIC determination showed that *triclabendazole* and its derivatives TCBZ-SO and TCBZ-SH possess high in vitro extracellular antimicrobial activity against methicillin-susceptible and methicillin-resistant staphylococci. Intracellularly, triclabendazole demonstrated a bactericidal effect at 32 μg/mL (16 x MIC). Conversely, in a mouse model of systemic *S. aureus* infection, repeated four-hourly oral treatment with triclabendazole demonstrated significant bacteriostatic activity [25].

Antibacterial effects have been reported for **antitumoral chemotherapeutic agents**. As shown in Table 1, doxorubicin, bleomycin, and gemcitabine exhibited higher antimicrobial effects than cytarabine, cyclophosphamide, methotrexate, and 5-fluorouracil against *S. pseudintermedius*. However, methotrexate and 5-fluorouracil caused modifications of bacterial growth by decreasing colony size [29].

**The thiazole platform** has attracted attention recently due to its multiple therapeutic potential including anticonvulsant, antitumoral, and antimicrobial activity. Thiazole compounds 1–6 were proposed by Mohammad et al. as antimicrobial agents against canine *S. pseudintermedius* [30]. They showed potent antibacterial activity against both methicillin-susceptible and resistant clinical isolates (Table 1). Thiazole compounds were demonstrated to be bactericidal at different time-killing (min. 2 h, max. 12 h) and to possess a post-antibiotic effect of at least 8 h. Compounds 3, 4, and 5 exhibited the best safety profile. Most importantly, all six compounds were able to restore the susceptibility of MRSP to oxacillin, suggesting their use as antimicrobial resistance breakers in combination with β-lactam antibiotics. Their bactericidal activity was associated with unknown mechanisms other than cellular membrane disruption.
vetsci-11-00311-t001_Table 1Table 1Lists of drugs purposed and repurposed for the treatment of animal dermato-pathogenic *Staphylococcus* spp.DrugMIC Range(μg/mL)Target TestedReferenceFosmidomycin25.9 (μM)SP[20]Leopolic-acid-inspired compound8MSSP[23]32MRSPAntiparasitic drugsLasalocid0.5–1MRSA[26]Monensin2–8Salinomycin0.5–2Narasin0.125–0.5
0.06–0.25MSSP, MRSP[27]Oxyclozanide0.5–1MSSP[24]0.5–2MRSP1SA ATCC 29213Triclabendazole2–4MSSA, MRSA, MSSP, MRSP[25]TCBZ-SO ^a^8MSSA, MRSATCBZ-SO_2_ ^a^8MSSA >256MRSATCBZ-SH ^a^2MSSA 4MRSATCBZ-OH ^a^>256MSSA, MRSAChemotherapeutic agentsBleomycin2–64SP[29]Doxorubicin2–4Cytosine arabinoside>64Cyclophosphamide>64Methotrexate>645-Fluorouracil>64Gemcitabine8–64Thiazole compounds10.35–0.69MSSP, MRSP[30]20.30–0.4630.71–0.4840.73–1.4750.4260.80–0.40^a^ Triclabendazole derivatives. Abbreviations: SP: *Staphylococcus pseudintermedius*; MSSP: Methicillin-susceptible *S. pseudintermedius*; MRSP: Methicillin-resistant *S. pseudintermedius*; MSSA: Methicillin-susceptible *Staphylococcus aureus*; MRSA: Methicillin-resistant *S. aureus*.


## 4. Natural Products

Natural compounds and plant-based antimicrobial extracts have emerged as valuable alternatives to conventional antibiotics and more than a thousand studies have been published in this field per year [31]. In particular, the in vitro antimicrobial activity of natural products against *Staphylococcus* species has been widely explored in the last five years, and the results are summarized in Table 2.

Many essential oils (EOs) thanks to their bioactive chemical compounds can be used in canine skin disorders. In 2024, the antibacterial activity of volatile oil from *Atractylodis rhizoma* (VOAR) was assessed against 30 *S. pseudintermedius* clinical isolates in vitro [32]. The MICs (total volatile oil content) ranged from 0.00315% to 0.00625%, except for one strain where the MIC was 0.0125%. The authors found no significant difference in MICs of VOAR between multidrug-resistant and non-multidrug-resistant bacteria. In the same study, they also evaluated VOAR therapeutic efficacy for canine superficial pyoderma using a murine model in vivo. Interestingly, they found that VOAR could effectively play a role in controlling *S. pseudintermedius* load on the body surface of mice and improve skin infection through an additional anti-inflammatory effect. *Cinnamomum zeylanicum* essential oil (CZEO) was tested on several *Staphylococcus* species isolated from canine otological infections, showing the inhibition of bacterial growth at a concentration ranging between 0.5–1 mg/mL [33]. Another study on 11 different essential oils against both MRSP and MSSP isolated from canine pyoderma, confirmed the efficacy of *C. zeylanicum* EO, with a MIC value of 1:1024 *v*/*v* for all tested isolates [34]. Four EOs, namely *Rosmarinus officinalis* (RO), *Juniperus communis* (GI), *Citrus sinensis* (AR), and *Abies alba* (AB) were tested against *S. pseudintermedius* with MIC values ranging from 1:32 to 1:2048 in dependence of the compound (Table 2). Interestingly, the same authors also tested these four EOs associated with silver nanoparticles (AgNPs), demonstrating that the incorporation of AgNPs allows a decrease in the values of the MIC for each EO examined [35]. Antimicrobial in vitro efficacy of oregano oil, thyme oil, and their main phenolic constituents against *S. pseudintermedius* isolates associated with canine otitis externa showed MIC values ranging from 100 to 292 µg/mL [36]. Regarding the antibacterial activity of manuka EO, there is little data available; however, some researchers reported good antimicrobial activity of manuka EO (MIC ranging from 0.125 to 128 µg/mL) against *S. pseudintermedius* isolated from canine pyoderma and otitis samples [37]. Manuka honey essential oil hydrogel has also been tested in vivo in full-thickness wounds in dogs, but the results of this study did not provide evidence to support its clinical application in small, acute wounds in healthy dogs [38].

In recent years, the antibacterial properties of *Allium* (*Allium sativum L.* and *Allium cepa L*.) have been extensively studied, including their efficacy against multidrug-resistant bacteria. In particular, in vitro antimicrobial activity of two *Allium*-derived compounds, propyl propane thiosulfate (PTS) and propyl propane thiosulfonate (PTSO) was evaluated against 30 multi-resistant *S. pseudintermedius* isolated from dogs [39]. The probability of clinical success for each compound was predicted by establishing a cut-off point of 62.5 µg/mL. Consequently, MICs above this value were considered resistant (13 strains), whereas those equal to or lower than this value were considered sensitive (17 strains). Mangosteen (*Garcinia mangostana* Linn) is a tropical fruit tree cultivated in South Asia, and its bioactive secondary metabolites are xanthone derivatives with strong pharmacological activity. α-mangostin (α-MG) is a major xanthone derivative that exhibits antibacterial effects. Park et al., investigated the antimicrobial activity of α-MG against clinical isolates of *Staphylococcus* species from skin diseases of dogs and cats and also evaluated α-MG therapeutic potential in a murine model [40]. MICs of α-MG ranged from 1 to 16 µg/mL. Interestingly, they also demonstrated that ultrastructural bacterial alterations are the main action mechanisms of α-MG in killing *S. pseudintermedius* [40], comparable to the results from *S. aureus* treated with antimicrobial peptides [41]. α-MG significantly reduced the bacterial load, partially restored the epidermal barrier, and suppressed the expression of pro-inflammatory cytokine genes in skin lesions induced by *S. pseudintermedius* in a murine model. Slightly different results were obtained in a previous study on α-MG antimicrobial activity against 23 *S. pseudintermedius* isolates from canine pyoderma (11 MRSP and 12 MSSP) with MIC values ranging from 0.49 µg/mL to 7.8 µg/mL [42].

The antimicrobial activity of *Piper betle* L against 20 clinical isolates of *S. pseudintermedius* (10 MSSP and 10 MRSP) was tested through the Kirby-Bauer disk diffusion method using *P. betle* extract disks (250, 2500, and 5000 μg). MIC values against all isolates were 250 μg/mL [43]. Several studies have described the antimicrobial activity of different extracts from the leaves of *Psidium guajava*, including the activity of methanolic extracts against methicillin-resistant *S. aureus* (MRSA) [44]. Broth microdilution assay demonstrated that *P. guajava* inhibited microbial growth at a concentration of 6.8 mg/mL for all reference strains tested (*S. aureus* and *S. pseudintermedius*). The same authors also tested *P. guajava* through the disk diffusion test [43]. However, it should be mentioned that studies on the antimicrobial activity of plant extracts based on agar diffusion have limited value [31].

Plant by-products usually lack economic value, and are managed as waste; however, several works have recently explored their potential valorization in a circular economy model perspective. Antimicrobial activity has been evaluated in pomegranate, quince, persimmon leaf, peel, and seed [45]. Leaves showed higher concentrations of phenolics than the peel and seeds of fruits. When comparing the three fruits, pomegranate leaf and peel extracts showed the highest antimicrobial activity (10–25 mg/mL), followed by the quince and persimmon leaf extract (25–50 mg/mL).

Antimicrobial compounds derived from probiotics, such as bacteriocins, are promising alternatives to conventional antibiotics. The antibacterial activity of one of the main bacteria in canine fecal microbiota, *Ligilactobacillus animalis*, was evaluated in vitro and in vivo [46]. It significantly inhibits the growth of multidrug-resistant (MDR) *S. pseudintermedius*, suggesting it could be used as a potential alternative to classic antibiotics for staphylococcal infections in dogs. Similarly, harzianic acid, the main metabolite produced by fungal extraction, showed good antimicrobial activity against MDR-*S. pseudintermedius* strains [47].

Natural substances may successfully be an alternative to synthetic compounds. However, more efforts should be made to manage the lack of standardization in measuring the antimicrobial activity of plant-based products and set up the proper assays to confirm their activity to obtain comparable and trusty results.
vetsci-11-00311-t002_Table 2Table 2In vitro antimicrobial activity of plant-based compounds for the treatment of animal dermato-pathogenic *Staphylococcus* species.Natural CompoundMIC RangeTarget TestedReference*Atractylodis rhizoma*0.00315–0.0125%SP[32]*Allium species*≥62.5 µg/mLMRSP[39]α-Mangostin1–8 µg/mLSP[40]1–8 µg/mLSS2–8 µg/mLSF1–16 µg/mLSE4–16 µg/mLSA
0.49–7.8 µg/mLMRSP, MSSP[42]*Piper betle*250 µg/mLMRSP, MSSP[43]*Psidium guajava*6.8 mg/mLSP[45]
6.8 mg/mLMRSA (ATCC 43300)Pomegranate10–25 mg/mLSP, SA[48]Quince25–50 mg/mLSP, SAPersimmon25–75 mg/mLSP, SA*Cinnamomum zeylanicum*0.5 mg/mLSP[33]0.5 mg/mLSSc1 mg/mLSS1:1024 *v*/*v*MRSP, MSSP[34]*Melissa officinalis*1:512–1:1024 *v*/*v*MRSP, MSSP*Leptospermum scoparium*1:512–1:1024 *v*/*v*MRSP, MSSP*Satureja montana*1:512–1:1024 *v*/*v*MRSP, MSSPOregano 140–281 µg/mLMRSP, MSSP[36]Carvacrol146–292 µg/mLMRSP, MSSPThyme137–275 µg/mLMRSP, MSSPThymol100–200 µg/mLMRSP, MSSP*Rosmarinus officinalis*1:64–1:256MRSP[35]*Juniperus communis*1:32–1:512MRSP*Citrus sinensis*1:64–1:2048MRSP*Abies alba*1:64–1:2048MRSPManuka oil2^−9^ to 2^−6^%, *v*/*v*MRSP, MSSP[37]Abbreviations: SP: *Staphylococcus pseudintermedius*; MSSP: Methicillin-susceptible *S. pseudintermedius*; MRSP: Methicillin-resistant *S. pseudintermedius*; SA: *Staphylococcus aureus*; MRSA: Methicillin-resistant *S. aureus*; SE: *Staphylococcus epidermidis;* SSc: *Staphylococcus schleiferi*; SS: *Staphylococcus simulans.*


## 5. Metal Nanomaterials

A rapidly growing field is nanotechnology, aiming to synthesize and characterize nanoparticles (NPs) with several applications, including their possible use in dermatology [49]. In this scenario, silver nanoparticles (AgNPs) demonstrated peculiar characteristics including antimicrobial action and it has been shown that the different surface charges of their coatings can affect the physical interaction of AgNPs with microorganisms [50]. AgNPs were biosynthesized from an infusion of *Curcuma longa* (ClAgNPs) and the culture supernatant of *E. coli* (EcAgNPs) and their antibacterial properties were evaluated alone and in combination with antibiotics. EcAgNPs alone showed the highest antibacterial activities, resulting in MIC values ranging from 0.438 ± 0.18 µM (*P. aeruginosa*) to 3.75 ± 3.65 µM (*S. pseudintermedius*) compared to those of ClAgNPs: 71.8 ± 0 µM (*P. aeruginosa*) and 143.7 ± 0 µM (*S. pseudintermedius*) [51]. AgNPs also displayed a significant dose-dependent antibiofilm activity and reduced *S. pseudintermedius* biofilm formation at concentrations of 20 and 10 μg/mL [52]. The anisotropic AgNPs showed antimicrobial efficacy against *S. pseudintermedius* (MIC range: 2–64 µg/mL) with low cytotoxicity to human cell lines [53] and reduced scar formation on wounds contaminated with MRSP in a mice model [54]. An alternative to AgNPs to treat MDR-staphylococci is represented by nanosulfur. It has been recently demonstrated that nanosulfur is effective against clinical isolates of MDR-*S. pseduintermedius* in both planktonic and biofilm states [55].

The incomplete ferrous salt (II, III) of polyacrylic acid, also known as IIS-PAA, feracrylum, or PAA, is a water-soluble hydrophilic polymer product with hemostatic and wound-healing properties. Solc et al. demonstrated the in vitro antibacterial effect of IIS-PAA against MRSP and MRSA, suggesting a potential clinical use in veterinary dermatology [56]. In iron pathways, DIBI is a novel water-soluble hydroxypyridinone-containing iron-chelating agent that deprives bacteria of growth-essential iron and has been previously shown to inhibit MRSA [57]. All *S. pseudintermedius* strains showed high susceptibility to DIBI (MIC 2 µg/mL), suggesting that it could be a potential non-antibiotic alternative for canine *S. pseudintermedius* otitis [34]. Interestingly, ferric hydroxamate uptake protein D (FhuD), which is implicated in bacterial iron uptake has been characterized. The authors demonstrated that FhuD is involved in the capture of iron (III) hydroxamates, which is likely to be crucial during the bacterial life cycle, making it a potential therapeutic target for *S. pseudintermedius* infections [58]. Finally, gallium, a semi-metallic element, exhibits antimicrobial activity by substituting important iron-dependent bacterial pathways. Arnold et al. tested the inhibitory activity of gallium maltolate (GaM) against veterinary clinical isolates of 34 MRSA and 88 MRSP. The MIC values for GaM against MRSP (range: 0.25 to >4 mg/mL) were significantly greater than those for MRSA (range: 0.0625 to >4 mg/mL) [59].

## 6. Skin Antiseptics

Topic chemical antiseptics are widely used to treat cutaneous infections to reduce the administration of antibiotics in small animal clinical practice. There are several formulations of topical antiseptics available based on the location of affected areas. As a general rule, formulations such as shampoo and spray are recommended for extensive or generalized pyoderma, while topical ointments and wipes can be used for the treatment of localized infections and otitis. General guidelines for the topical antimicrobial treatment of superficial bacterial folliculitis are described in the literature [5]. Below, we discussed the active ingredients specifically proposed for the treatment of dermato-pathogenic *Staphylococcus* spp. in companion animals.

(a)**Chlorhexidine,** a bisbiguanide compound, is the most used antiseptic in topic formulations for companion animals in virtue of its high antimicrobial activity against *Staphylococcus* spp. (MIC range: 0.5–4 μg/mL) [60,61]. An important question associated with the use of chlorhexidine is whether it can be used as monotherapy or in combination with antibiotics. Some authors reported clinical improvement of canine superficial pyoderma treated with chlorhexidine acetate or gluconate alone, but the clinical resolution was hard to achieve [62]. By comparing topical chlorhexidine to systemic amoxicillin–clavulanic acid, Borio and colleagues demonstrated that the treatment with 4% chlorhexidine digluconate shampoo (twice weekly) and solution (once daily) for 4 weeks had comparable efficacy to that of the systemic antibiotic, resulting in clinical resolution of superficial pyoderma in all dogs including those infected by MRSP [63]. However, the combination of chlorhexidine with certain antimicrobial agents may provide useful synergistic interactions, such as chlorhexidine/miconazole [61].(b)**Polyhexanide**, a polyhexamethylene biguanide compound, is a broad-spectrum antiseptic with a low-risk profile that acts on bacterial membranes affecting its integrity. It was suggested that the microbicidal efficacy of polyhexanide is comparable to that of chlorhexidine against dermato-pathogenic canine isolates [64], but polyhexanide may have higher safety because it does not contain the toxic terminal chlorobenzene substituents.(c)**Olanexidine gluconate**, a novel biguanide antiseptic, was developed in 2015 for human use as a skin antiseptic. It demonstrated a wide bactericidal activity comparable to that of chlorhexidine, more markedly against Gram-positive bacteria including MRSA and MRSP (MIC 0.23 μg/mL) [65,66]. In a randomized, single-blinded controlled clinical trial, olanexidine spray administered topically for 10 days substantially improved clinical signs in dogs with atopic dermatitis and superficial pyoderma caused by MSSP and MRSP, and its effect was comparable to bathing once a week with chlorhexidine [66].(d)**Sodium hypochlorite** was proposed as a topical antiseptic to treat canine staphylococcal pyoderma. It showed an overall in vitro MBC of 1:32 for canine skin-isolated MRSP strains (range 1:32–1:1024) [67]. Other studies suggested that the bactericidal concentrations of sodium hypochlorite against *S. aureus* and *S. pseudintermedius* range from 0.05 to 0.005% [66]. At these concentrations, sodium hypochlorite seems also to reduce the pro-inflammatory response [68]. The major issue related to the use of sodium hypochlorite is balancing efficacy and tolerability. A single spray application of 0.05% sodium hypochlorite was demonstrated to reduce the bacterial load within 20 min on the skin of dogs without signs of irritation; however, 0.05% sodium hypochlorite significantly reduced the percentage of viable keratinocytes in vitro [68]. Additionally, a change in skin microbiome with an increase of coagulase-negative staphylococci was observed in the presence of sodium hypochlorite [68]. On the other hand, repeated 0.005% hypochlorite bleach baths over four weeks (twice weekly for 15 min) were well tolerated in healthy dogs without significant changes in the density of *S. pseudintermedius* [69].(e)**Oxychlorosene**, an organic complex of dodecylbenzenesulfonic sodium salt and hypochlorous acid, is commonly used as a topical antiseptic to treat localized human infections. Despite, to our knowledge, no study investigated the in vivo safety and the efficacy of sodium oxychlorosene on canine and feline skin infections, in vitro time-kill assays support its use to treat *S. pseudintermedius* infections. 0.2 and 0.4% sodium oxychlorosene solutions showed a rapid bactericidal effect with greater than 99.9% average *S. pseudintermedius* reduction within 5 s [70].(f)**Fatty acids** have been proposed as topical antimicrobials for the treatment of dermato-pathogenic *Staphylococcus* spp. [71]. Among these, oleic acid salt and linoleic acid salt in ultrapure soft water (UPSW), in which Ca++ and Mg++ were replaced with Na+, exhibited strong in vitro antibacterial activity against *S. aureus*, *S. intermedius*, and *S. pseudintermedius*. In vivo, shampoo treatment with liquid soap containing 10% linoleic acid in UPSW was demonstrated to improve spontaneous dermatitis in dogs by significantly decreasing transepidermal water loss, pruritus, and alopecia.

Overall, topical antiseptics have the potential to be used as an alternative to antibiotics to treat localized skin infections in companion animals. However, randomized and blinded in vivo studies are needed to estimate the exposure times and the concentrations of active ingredients achieved at the application site in animals with different hair coats testing various antiseptic formulations.

### Physical Agents

Physical antiseptics were also tested against *S. pseudintermedius* and *S. aureus*. In particular, irradiation with blue light **phototherapy** at 465 nm completely inhibited the growth of MRSA colonies at the dose of 112.5 (30 min exposure) and 225 J/cm^2^ (90 min exposure). On the contrary, the percentage reduction of the colony counts for MSSP and MRSP was only 11.7% and 21.2%, respectively, at the highest dose tested (225 J/cm^2^) [72]. Exposure to blue light for 90 min using a photosensitizer, such as δ-aminolevulinic acid was shown to improve the bactericidal activity against *S. pseudintermedius* [73]. Moreover, when applied daily for 2 months, blue light at lower absorbance (308 nm) was able to reduce the load of *S. pseudintermedius* in allergic canine skin [74]. In vivo, phototherapy at 810 nm in combination with indocyanine green as photosensitizer applied four times for 30 s with a dosage of 50 J/cm^2^ showed effectiveness in treating para-aural abscess caused by multi-resistant *S. pseudintermedius* [75]. Overall, both blue and red light have shown promising results for the treatment of staphylococcal skin infections, even if the infrared spectrum may allow a deeper penetration in tissues.

**Fluorescent light energy (FLE)**, a type of phototherapy that is based on the translation of light energy into a low-energy emission of fluorescence, did not have a bactericidal effect on *S. pseudintermedius* and *S. aureus* in vitro [76]. On the contrary, dogs with superficial and deep pyoderma and interdigital furunculosis caused by multidrug-resistant bacteria, including *S. pseudintermedius* and *S. aureus,* showed a decrease in lesion scores when treated by FLE twice weekly without any adjunct therapy [77,78].

Additionally, when used in conjunction with systemic antibiotics, FLE was shown to reduce the time to achieve clinical resolution compared to dogs receiving only antibiotics. This led to the reduction of the length of antibiotic treatment in cases of interdigital furunculosis and deep pyoderma in dogs [79,80]. 

**Cold atmospheric plasma (CAP),** a partially ionized gas, is considered a last-generation therapy for bacterial infections based on its ability to disrupt the bacterial cell wall and cause DNA damage. Based on the production method, CAP is distinguished into different types, including cold atmospheric microwave plasma (CAMP). It was demonstrated that CAMP has a time-dependent activity against staphylococci [81]. In particular, at a plasma intensity of 30 W, the survival rates of *S. aureus* were 65%, 28.1%, and 9.9% when irradiated for 10 s, 30 s, and 60 s, respectively. At 50 W, the survival rates were 50.3%, 18%, and 2.8% at 10 s, 30 s, and 60 s of exposure. *S. pseudintermedius* survival rates were similar to those of *S. aureus*. CAMP bactericidal activity against *Staphylococcus* was not connected to the antimicrobial resistance profile. Overall, CAMP appeared to be less effective against staphylococci than Gram-negative bacteria.

## 7. Antimicrobial Peptides

There exists a considerable body of literature on antimicrobial peptides (AMPs) as bioactives against cutaneous staphylococcal infections. AMPs are small cationic molecules naturally produced by both prokaryotic and eukaryotic cells that act as defense factors in the innate immune response. Table 3 summarizes the AMPs proposed for the treatment of skin infections caused by *Staphylococcus* spp. in companion animals.

Among anti-staphylococcal AMPS, defensins are the most studied peptides. These are secreted from cells of different hosts and showed variable antimicrobial activity with NZ2114, a variant of the fungal defensin plectasin, canine β-defensin 103 and its variant CBD103ΔG23 and human β-defensin 3 being the most effective against MSSP and MRSP [82,83,84]. A synergistic/additive activity of β-defensin 103 and peptide cathelicidin in combination with chlorhexidine was also suggested showing a MIC value of 100 μg/mL against MSSA, MRSA compared to 25–5 μg/mL against MSSP, MRSP [85]. It is worth mentioning that NZ2114 formulated as a spray at 5 mg/mL in combination with 5% N-methylpyrrolidone and 10% propylene glycol significantly reduced the *S. pseudintermedius* load in the skin lesions of a mouse model of superficial pyoderma [84].

Nisin produced by *Lactococcus lactis* is a lantibiotic peptide known by virtue of its wide antibiotic target. Over the years, varied nisin-derivatives were developed, including Nisin I4V with enhanced antimicrobial activity against *S. pseudintermedius* [86]. Nisin I4V was also demonstrated to inhibit *S. pseudintermedius* biofilm formation and reduce the biomass of preformed biofilms.

Another large family of AMPs includes temporins isolated from the skin of *Rana temporaria*. Peptide 8, a derivative of temporin-L, showed high antimicrobial activity when tested alone and synergism in combination with oxacillin. Despite that peptide 8 was unable to inhibit biofilm formation at 0.78 μM, it caused a significant decrease in MSSP and MRSP biofilm viability [87].

Tang and colleagues studied the antimicrobial activity of allomyrinasin, andricin B, pinipesin, nigrocin-HLM, and Hs02 against *Staphylococcus* spp. isolates from the skin of dogs with pyoderma [88]. Allomyrinasin and andricin B exhibited considerable antimicrobial activity against *S. pseudintermedius* (Table 3) and rapid bactericidal effect (time for total killing at 1 xMIC = 300 min). It was supposed that the anti-staphylococcal mechanism primed by these AMPs is linked to membrane lysis. Allomyrinasin showed the best inhibiting and eradicating activity against staphylococcal biofilm (Minimum Biofilm Inhibitory Concentration (MBIC) = 32 μg/mL; Minimum biofilm eradication concentration (MBEC) = 128 μg/mL) and alleviated the inflammation response by decreasing IL-6. Synergism of the combination allomyrinasin/amoxicillin was also observed. The efficacy of allomyrinasin and andricin B was also confirmed in an in vivo mouse model of *S. pseudintermedius* skin infection.

The AMPs aurein 1.2, CAMEL, citropin 1.1, protegrin-1, pexiganan, temporin A, and uperin 3.6 were also tested against staphylococcal pathogens associated with canine pyoderma [89]. All tested peptides were active against all *S. aureus* and *S. pseudintermedius* strains tested, with median MICs ranging from 2 μg/mL (Uperin 3.6, CAMEL, Protegrin-1) to 128 μg/mL (Aurein 1.2). Overall, these AMPs appeared to be more effective against *S. pseudintermedius* than *S. aureus*.

Despite the interest generated by the antimicrobial properties of AMPs, some issues limited their practical application, principally the low stability as a result of the susceptibility of AMPs to enzymatic degradation. To overcome these challenges, chemical and structural modifications have been generated to enhance the stability of AMPs. In particular, peptoids, which are polymers of N-substituted alkylglycines structurally analogous to peptides, have been introduced. Greco et al. characterized a peptide-peptoid hybrid (B1) and a peptoid (D2) showing MIC of 2–4 and 1–2 μg/m against *S. pseudintermedius*, respectively [90]. These peptides analogs were suggested as antimicrobials for the local treatment of canine superficial pyoderma in an oil-in-water cream formulation. The authors also performed structure–activity relationship analyses of D2 resulting in 19 peptide/peptoid analogs. Among these, the derived compound D2D exhibited high efficacy against *S. pseudintermedius* and other pathogens of canine pyoderma [91]. It is noteworthy to mention the peptidomimetic fluorinated α-peptide/β-peptoid hybrids derived by fluorination and N-terminal end-group modifications of α-peptide/β-peptoid hybrids [92]. These modifications contributed to enhancing the antimicrobial activity against methicillin-susceptible and methicillin-resistant *S. aureus* and *S. pseudintermedius*; however, a major issue associated with the joint use of such modifications was the increase of the hemolytic effect. A rigorous design and optimization of peptidomimetics is therefore needed.
vetsci-11-00311-t003_Table 3Table 3Antimicrobial peptides and peptidomimetics proposed for the treatment of staphylococcal cutaneous infections in companion animals.Antimicrobial PeptideOriginMIC Range(μg/mL)Target TestedReferenceβ-defensin 103Canine skin epithelium2–5MSSP, MRSP[82]25–5MSSP, MRSP[85]100MSSA, MRSA[85]β-defensin 3Human skin epithelium3.125–25SP[83]β-defensins 6, 12Avian leukocytes and epithelial cells16–256MRSP,SA ATCC 29213[93]CathelicidinCanine leukocytes and epithelial cells25–50MSSP, MRSP[94]100–200MSSA, MRSANZ2114Variant of fungal defensin plectasin1–2SP, MDR-SP[78]Nisin I4VSemisynthetic0.25–1SP[86]1.5SATemporin-L-derived peptide 8Semisynthetic1.56MSSP[87]6.25MRSPAllomyrinasinAllomyrina dichotoma8–32SP[88]32SC>256SH, SS128SSiAndricin BAndrias davidianus32–128SP64SC≥256SH, SS, SSiPinipesinScolopendra subspinipes64–256SP>256SC, SH, SS128SSiNigrocin-HLMSynthetic32–256SP128SC64SH>256SS16SSiHs02Intragenic32–256SP32SC128SH>256SS64SSiUperin 3.6Litoria genus2–4SP[88]8–16SACAMELSynthetic2–4SP4SAProtegrin-1Porcine leukocytes1–8SP4–32SAPexigananSynthetic1–8SP4–8SACitropin 1.1Litoria genus2–8SP8–16SATemporin ARana temporaria4–8SP8SAAurein 1.2Litoria genus32–64SP128–256SAPeptide-peptoid hybrid B1Synthetic2–4MSSP, MRSP[90,95]Peptoid D2Synthetic1–2MSSP, MRSP[90,91]Fluorinated α-peptide/β-peptoid hybridsSynthetic0.5–32MSSP, MRSP[92]2–32MSSA, MRSAAbbreviations: SP; *Staphylococcus pseudintermedius*; MSSP, Methicillin-susceptible *S. pseudintermedius*; MRSP, Methicillin-resistant *S. pseudintermedius*; MDR, Multidrug-resistant; SA, *Staphylococcus aureus*; MSSA, Methicillin-susceptible *S. aureus*; MRSA, Methicillin-resistant *S. aureus*; SC, *Staphylococcus cohnii*; SH, *Staphylococcus haemolyticus*; SS, *Staphylococcus sciuri*; SSi, *Staphylococcus simulans.*


## 8. Bacteriophages

By their ability to target specific bacteria, including antimicrobial-resistant pathogens, bacteriophages may be used as an alternative to conventional antibiotic therapy. Previous studies have reported bacteriophages with preferential activity against antibiotic-resistant staphylococcal isolates from canine cutaneous infections.

Phages belonging to the ***Siphoviridae*** family were isolated from canine feces using MRSP strains as initial hosts [96]. They were shown to have high lytic activity against MRSP and *S. schleiferi* and low efficacy against MSSP, while they had no lytic infectivity towards *S. aureus*, *S. epidermidis*, *S. warneri*, and *S. haemolyticus*. Environmental phages pSp-J and -S, clustering in *Phietavirus* genus included in the *Siphoviridae*, showed a significant effect in preventing biofilm formation and in degrading preformed biofilm mass of *S. pseudintermedius* in a concentration-dependent manner [97].

***Myoviridae*** family includes the staphylococcal phage named phiSA012 isolated from sewage influent [98]. It has little lytic activity against *S. pseudintermedius* and *S. schleiferi*. On the other hand, its endolysin Lys-phiSA012 showed rapid antimicrobial activity by significantly reducing *S. pseudintermedius* and *S. schleiferi* viable cells [99]. *Myovirus* phages from canine hair and skin are noteworthy because they revealed a strong polyvalent activity against both *S. pseudintermedius* and *S. aureus* [100], thus constituting promising candidates for anti-staphylococcal therapeutic cocktails.

## 9. Adjuvants in Antibiotic Treatments

Adjuvants are chemical molecules with weak or absent antimicrobial activity that can be used to enhance the efficacy of primary antibiotics. Below, we present potential adjuvants proposed for the combined treatment of dermato-pathogenic staphylococcal species:(a)**Inhibitors of metabolic pathways**. O-acetylserine sulfhydrylase (OASS) inhibitor is a blocker of the cysteine biosynthetic pathway. OASS was tested in combination with colistin against *S. aureus* and *S. pseudintermedius* [101]. This antibiotic has a very limited potency on *Staphylococcus* spp. that are intrinsically resistant to polymyxins [102]. When combined with OASS, the MIC of colistin decreases suggesting additive activity [101].(b)**Enzymes** with antimicrobial activity have been developed to treat bacteria that are resistant to common antibiotics and biocides, particularly those producing strong microbial biofilms. Known antimicrobial enzymes include the class of proteolytic and polysaccharide-degrading enzymes, such as lysostaphin and lysozyme, and oxidative enzymes that showed promising effects on human pathogenic staphylococci. Seminal reviews have been previously published on this topic [103]. Among the polysaccharide-degrading enzymes, dispersin B was proposed for the treatment of canine pyoderma caused by biofilm-producing *S. pseudintermedius*. DispersinB is considered an anti-biofilm enzyme that hydrolyzes poly-N-acetylglucosamine, a core component of biofilm. As a result, this enzyme affects the formation and stability of biofilm but it does not significantly impact the viability of staphylococcal cells; therefore, it should be considered as an adjunctive treatment.(c)**Efflux pump inhibitors (EPIs)**. Overexpression of membrane efflux pumps was demonstrated to be an effective resistance mechanism in *Staphylococcus* spp., including *S. pseudintermedius* and *S. aureus* [104,105]. Several molecules, known as antibiotic resistance breakers, were designed and synthesized to inactivate *S. aureus* efflux pumps, principally NorA [106]. Recently, some compounds previously reported as *S. aureus* NorA EPIs proved to have EPI activity against S. *pseudintermedius,* particularly the scaffolds 2-arylquinoline, nicardipine, and 2-phenyl-4-carboxy-quinoline [107]. Finally, high efflux appears to be linked to the downregulation of DNA repair and mutagenesis promotion [108,109]. Accordingly, EPI derivatives might be proposed as anti-evolutive drugs to prevent resistance evolution and preserve the efficacy of existing antibiotics.(d)**Miscellaneous adjuvants in otitis externa.** Other substances were proposed as potential adjuvants in otitis caused by S. *pseudintermedius*, including monolaurin, monocaprin, N-acetylcysteine (NAC), polymyxin B nonapeptide, Tris-EDTA, Tris-HCL and disodium EDTA [110]. Tested alone, they showed limited to no antimicrobial efficacy. However, synergy with antibiotics has to be investigated to understand their potential as staphylococcal adjuvants. Of note, NAC had an indifferent effect in combination with enrofloxacin against *S. pseudintermedius*, while the combination of NAC/gentamicin resulted in indifferent or antagonistic interactions [111]. The antibiofilm activity of NAC, Tris-EDTA, and disodium EDTA was also investigated [28]. MBEC of NAC ranged from 5000 to 10,000 μg/mL against biofilm produced by *S. pseudintermedius* and a significant decrease of *S. pseudintermedius* biofilm was observed at concentrations of 2500 μg/mL or above. No MBEC was achieved for Tris-EDTA and disodium EDTA against *S. pseudintermedius* biofilm, but a significant reduction in biofilm growth was observed at concentrations equal to or higher than 1500/470 μg/mL and 470 μg/mL, respectively.

In general, the use of adjuvants may lead to an enhancement of the antibiotic activity by reducing the clinical dosage and/or minimizing resistance. The emergence of resistance may indeed be limited by using compounds with different bacterial targets, including not essential factors for the survival of the bacterium. In this way, bacteria may be under minor selective pressure to develop resistance [112].

## 10. Vaccines and Inflammasome Inhibitors

The literature on vaccine development against dermato-pathogenic *Staphylococcus* spp. is less consistent. By whole-proteome characterization and serological proteome analysis, Couto and colleagues identified several vaccine-candidate antigens of *S. pseudintermedius* [113]. In particular, four were considered highly immunogenic: (a) **SpsD** located in the cell wall is a presumed pathogenesis-associated protein; (b) a **predicted lipoprotein** (unknown function and subcellular localization); (c) **AtpA** is a membrane ATP synthase that activates motility and active transport and supports staphylococcal growth and metabolism; (d) **enoyl-[acyl-carrier-protein] reductase (NADPH) FabI protein** located in the cellular membrane is an essential factor for staphylococcal fatty acid biosynthesis.

By using a similar approach, additional potential vaccine targets for *S. pseudintermedius* were prioritized [114]. This process led to modeling chimeric vaccine constructs based on the best final epitopes of two selected proteins (**N-acetylmuramyl-L-alanine amidase** and **FtsQ-type polypeptide-transport-associated domain-containing protein**) combined with linkers and adjuvant sequence to boost the immune response.

Pyoderma caused by staphylococcal species in companion animals, more frequently *S. pseudintermedius*, is one of the most common reasons for antibiotic prescription in the veterinary sector. This means that the design of effective vaccines to combat *S. pseudintermedius* infection is strongly advocated.

Finally, inflammasome inhibitors have emerged as promising therapeutic drugs in the course of noninfectious and infectious diseases [115]. Antibiotics alone may be ineffective in the treatment of bacterial keratitis; therefore, inflammasome inhibitors have been suggested as alternative or combined therapy. Particularly, MCC950 was proposed for the treatment of canine keratitis caused by *S. pseudintermedius*, a complex pathological condition involving severe inflammatory response and corneal fibrotic healing [116]. By inhibiting NLRP3 inflammasome, NF-κB signaling, and Wnt/β-catenin and PI3K/AKT pathways, MCC950 alleviated cellular inflammatory responses and cell damage primed by *S. pseudintermedius* infection.

## 11. Conclusions

In many countries, strict guidelines on the prudent use of antibiotics in veterinary medicine are in force to safeguard their effectiveness. Attention has been given to skin infections of companion animals because they are extremely common in small animal practices and often require antibiotic administration. Effective management of cutaneous infections in companion animals requires a prudent approach that includes (1) careful assessment of the necessity of administering antibiotics, (2) choice of the administration route preferring topical formulations according to the extension and severity of skin lesions, and (3) selection of the antibiotic compound based on the results of antibiograms and prudence principles of antibiotic use to preserve critically important antibiotics for human use. Additionally, sustainable practices should be promoted by advancing research on non-antibiotic therapies. In this context, reduced administration of systemic antibiotics, topical treatments using antiseptics, physical agents, or natural compounds is attracting considerable research efforts, as well as the use of antimicrobial peptides. On the other hand, alternative therapeutic approaches such as the use of antimicrobial resistance breakers, drug repurposing, and vaccine development remain briefly addressed in the literature, despite their promising impact on dermato-pathogenic antimicrobial-resistant *Staphylococcus* strains. Further experimental investigations on alternative therapeutic strategies are thus recommended to reduce microbial resistance among staphylococcal isolates and decrease the spread of infections caused by multidrug-resistant strains to improve patient outcomes both in veterinary and human medicine.

## Figures and Tables

**Figure 1 vetsci-11-00311-f001:**
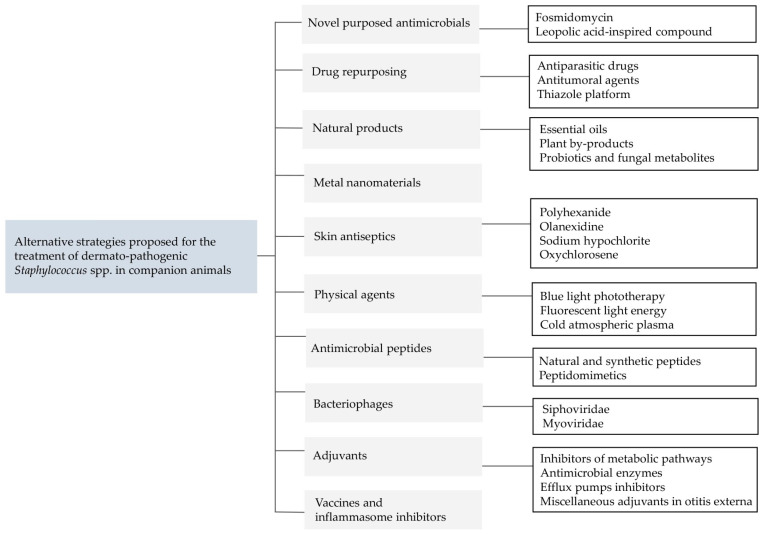
Summary of the alternative strategies proposed for the treatment of dermato-pathogenic *Staphylococcus* spp. in companion animals.

## Data Availability

No new data created.

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
