# Peer review of "Antimicrobial Strategies Proposed for the Treatment of *S. pseudintermedius* and Other Dermato-Pathogenic *Staphylococcus* spp. in Companion Animals: A Narrative Review"

_vetsci, 2024, doi:10.3390/vetsci11070311_

Round 1

Reviewer 1 Report

Comments and Suggestions for Authors

Authors are requested to discuss (with appropriate references) on how some of the antibiotics would be metabolized by companion animals and if it would be any different than what has been documented in case studies of humans and larger primates.

Additionally, if there is any data on pharmakokinetics of antibiotics that are currently prescribed by veterinarians. 

Comments on the Quality of English Language

Authors are suggested to review typographical errors found in the conclusion section of the review. 

Author Response

RESPONSE TO REVIEWERS

We thank the Reviewers for their time and effort in reviewing our manuscript. We believe it has been improved as a result of your comments. Please find below a point-by-point response to your comments and suggestions.

REVIEWER 1

Authors are requested to discuss (with appropriate references) on how some of the antibiotics would be metabolized by companion animals and if it would be any different than what has been documented in case studies of humans and larger primates.

Reply: We thank you for your suggestion. We added three references concerning the pharmacokinetics of clindamycin and fluoroquinolones in the context of the treatment of skin infections in companion animals. On the other hand, including data on how antibiotics are metabolized by companion animals and differences documented in case studies of humans and larger primates seems out of the scope of this review.

Additionally, if there is any data on pharmakokinetics of antibiotics that are currently prescribed by veterinarians.

Reply: Several publications on pharmacokinetics profiles of traditional antibiotics used in companion animals are available in the literature. However, it seems out of the scope of this review. In the revised version, we mention in the paragraph “Conventional antibiotics” that existing data on suggested doses for skin infections and therapeutic regimens is reported in a previous review and we refer to that (reference 5). Additionally, we added three references concerning the pharmacokinetics of clindamycin and fluoroquinolones in the context of the treatment of skin infections in companion animals.

Comments on the Quality of English Language

Authors are suggested to review typographical errors found in the conclusion section of the review.

Reply: The conclusion has been reviewed, thank you for your suggestion.

Reviewer 2 Report

Comments and Suggestions for Authors

I would like to thank the authors for this comprehensive review article and narrative review on current strategies to treat staphylococcus infections. The article and references are complete and interesting. However, I have a few suggestions and improvements:

  1. Figure 1:

    • To be more precise, the schema should be modified. Physical therapy should be a separate category and not included under skin antiseptics. This category should include Cold Atmospheric Plasma (CAP), blue light therapy, and fluorescent light energy.
  2. Textual Revisions:

    • The text should also reflect this categorization by including CAP, blue light therapy, and fluorescent light energy under a distinct physical therapy section.
  3. MoA and literature review:

    • The paragraph explaining the mechanisms of action and the results of studies on effectiveness should be expanded to provide a more extensive literature review. For instance FLE has been shown to be effective as a sole management therapy in superficial pyoderma.  (Front Vet Sci. 2023 Jun 2;10:1155105. doi: 10.3389/fvets.2023.1155105. eCollection 2023.)
    • Additionally, when used in conjunction with systemic antibiotics, FLE has been shown to reduce the time to achieve clinical resolution compared to dogs receiving only antibiotics. This reduces the length of antibiotic treatment in cases of interdigital furunculosis and deep pyoderma in dogs. 
      • Marchegiani, A., Spaterna, A., Cerquetella, M., Tambella, A.M., Fruganti, A., and Paterson, S. (2019). Fluorescence biomodulation in the management of canine interdigital pyoderma cases: a prospective, single-blinded, randomized, and controlled clinical study. Vet Dermatol, 30: 371-e109. https://doi.org/10.1111/vde.12785
      • Marchegiani A, Fruganti A, Spaterna A, Cerquetella M, Tambella AM, Paterson S. The Effectiveness of Fluorescent Light Energy as Adjunct Therapy in Canine Deep Pyoderma: A Randomized Clinical Trial. Vet Med Int. 2021 Jan 9;2021:6643416. doi: 10.1155/2021/6643416.
  4. Conclusion:

    • The conclusion can be expanded and improved by providing more information on antimicrobial stewardship. This should include detailed discussions on the risk of bacterial resistance and its implications for both veterinary and human health.
Comments on the Quality of English Language

Minor editing and spelling check

Author Response

RESPONSE TO REVIEWERS

We thank the Reviewers for their time and effort in reviewing our manuscript. We believe it has been improved as a result of your comments. Please find below a point-by-point response to your comments and suggestions.

REVIEWER 2

I would like to thank the authors for this comprehensive review article and narrative review on current strategies to treat staphylococcus infections. The article and references are complete and interesting. However, I have a few suggestions and improvements:

  1. Figure 1:
    • To be more precise, the schema should be modified. Physical therapy should be a separate category and not included under skin antiseptics. This category should include Cold Atmospheric Plasma (CAP), blue light therapy, and fluorescent light energy.

Reply: Figure 1 was modified according to the reviewer's suggestion.

  1. Textual Revisions:
    • The text should also reflect this categorization by including CAP, blue light therapy, and fluorescent light energy under a distinct physical therapy section.

Reply: Thank you, we agree. CAP was incorporated among physical therapies.

  1. MoA and literature review:
    • The paragraph explaining the mechanisms of action and the results of studies on effectiveness should be expanded to provide a more extensive literature review. For instance FLE has been shown to be effective as a sole management therapy in superficial pyoderma.  (Front Vet Sci. 2023 Jun 2;10:1155105. doi: 10.3389/fvets.2023.1155105. eCollection 2023.)
    • Additionally, when used in conjunction with systemic antibiotics, FLE has been shown to reduce the time to achieve clinical resolution compared to dogs receiving only antibiotics. This reduces the length of antibiotic treatment in cases of interdigital furunculosis and deep pyoderma in dogs. 
      • Marchegiani, A., Spaterna, A., Cerquetella, M., Tambella, A.M., Fruganti, A., and Paterson, S. (2019). Fluorescence biomodulation in the management of canine interdigital pyoderma cases: a prospective, single-blinded, randomized, and controlled clinical study. Vet Dermatol, 30: 371-e109. https://doi.org/10.1111/vde.12785
      • Marchegiani A, Fruganti A, Spaterna A, Cerquetella M, Tambella AM, Paterson S. The Effectiveness of Fluorescent Light Energy as Adjunct Therapy in Canine Deep Pyoderma: A Randomized Clinical Trial. Vet Med Int. 2021 Jan 9;2021:6643416. doi: 10.1155/2021/6643416.

Reply: Thank you for these suggestions. These references have been incorporated in the text (lines 418). The mechanism of cold atmospheric plasma has been also explained.

  1. Conclusion:
    • The conclusion can be expanded and improved by providing more information on antimicrobial stewardship. This should include detailed discussions on the risk of bacterial resistance and its implications for both veterinary and human health.

Reply: We agree. Conclusions have been expanded as suggested by the reviewer (please see lines 597-613).

Reviewer 3 Report

Comments and Suggestions for Authors

This manuscript, by Valentina Stefanetti et al., "Antimicrobial strategies proposed for the treatment of dermato-pathogenic Staphylococcus spp. in companion animals: a narrative review "highlights the importance of a multifaceted approach in developing effective antimicrobial strategies for companion animals, aimed at mitigating drug resistance and improving the clinical management of staph skin infections. I am very interested in this review and believe it will be of interest to other readers of this publication. However, the subject in this review is mainly Staphylococcus pseudintermedius, so the title should be modified more specifically. At the same time, it is necessary to elaborate the incompatibility between S. pseudintermedius and other Staphylococcus spp skin diseases more precisely in this paper. In addition, there are too many keywords, which makes the central argument not prominent enough.

Author Response

RESPONSE TO REVIEWERS

We thank the Reviewers for their time and effort in reviewing our manuscript. We believe it has been improved as a result of your comments. Please find below a point-by-point response to your comments and suggestions.

REVIEWER 3

This manuscript, by Valentina Stefanetti et al., "Antimicrobial strategies proposed for the treatment of dermato-pathogenic Staphylococcus spp. in companion animals: a narrative review "highlights the importance of a multifaceted approach in developing effective antimicrobial strategies for companion animals, aimed at mitigating drug resistance and improving the clinical management of staph skin infections. I am very interested in this review and believe it will be of interest to other readers of this publication. However, the subject in this review is mainly Staphylococcus pseudintermedius, so the title should be modified more specifically.

Reply: The title has been modified as per reviewer's suggestion.

At the same time, it is necessary to elaborate the incompatibility between S. pseudintermedius and other Staphylococcus spp skin diseases more precisely in this paper.

Reply: We explained better the incompatibility between S. pseudintermedius and other Staphylococci when appropriate

In addition, there are too many keywords, which makes the central argument not prominent enough.

Reply: Some keywords were deleted as per reviewer's suggestion.
